# Gender disparities in application and admission to the medical residency program in Peru: A cross sectional study from 2016 to 2023

**Daniel Medina-Neira** [1], **Brenda Caira-Chuquineyra**[2], **Daniel Fernandez-Guzman** [3]*

**1** Facultad de Medicina Humana, Universidad Católica de Santa María, Arequipa, Perú, **2** Facultad de Medicina Humana, Universidad Nacional de San Agustín de Arequipa, Arequipa, Perú, **3** Carrera de Medicina Humana, Universidad Científica del Sur, Lima, Perú

\* danferguz@gmail.com

## Abstract

### Objective

To investigate gender disparities in applications and admissions to the medical residency programs in Peru, focusing on differences in application and admission proportions between male and female.

### Methods

We conducted a cross-sectional study to assess the proportions of female applicants and admissions to medical residency programs in Peru from 2016 to 2023. Bayesian multilevel linear models were employed, incorporating random intercepts and slopes by specialty to account for variability across specialties. This approach provided initial proportions of female in 2016 (intercepts) and annual percentage changes (beta coefficients) for each specialty. A multilevel Poisson regression model with robust variance was used to determine if being female was associated with higher admission frequency.

### Results

Of the 48,013 applicants, 48% were considered female applicants. Most specialties exhibited an increasing trend in female applicants (+0.2% to +2% annually), except for Family Medicine, Hematology, Pediatric Neurology, and Pathological Anatomy (-0.6%, -0.6%, -0.7%, and -0.9% annually, respectively). The specialties with the highest proportions of female admissions were in Physical Medicine and Rehabilitation (71.9%), Dermatology (71.2%), and Pathological Anatomy (71.2%). In contrast, the lowest proportions were observed in Neurosurgery (18.9%), Thoracic and Cardiovascular Surgery (17.7%), and Urology (15.6%). Declining trends in female admissions were noted in Family Medicine, Hematology, and Pathological Anatomy (-0.6%, -0.6%, and -0.8% annually, respectively). In addition, being female was associated with an 18% lower probability of admission to the medical residency program (prevalence ratio: 0.82; 95% CI: 0.78–0.85; p-value: <0.001).

**Data Availability Statement:** The data was obtained from the official website of the Consejo Nacional de Residentado Médico (CONAREME): https://www.conareme.org.pe/web/ The application

processes and admission information from previous years are accessible through the following links: https://www.conareme.org.pe/web/proceso-de-admision-2016.php https://www.conareme.org.pe/web/proceso-de-admision-2017.php https://www.conareme.org.pe/web/proceso-de-admision-2018.php https://www.conareme.org.pe/web/proceso-de-admision-2019.php https://www.conareme.org.pe/web/proceso-de-admision-2020.php https://www.conareme.org.pe/web/proceso-de-admision-2021.php https://www.conareme.org.pe/web/proceso-de-admision-2022.php https://www.conareme.org.pe/web/proceso-de-admision-2023.php.

**Funding:** The author(s) received no specific funding for this work.

**Competing interests:** The authors have declared that no competing interests exist.

## Conclusion

This study identified persistent gender disparities in medical residency programs in Peru, with female applicants facing reduced probabilities of admission and exhibiting specialty-specific trends from 2016 to 2023.

## Introduction

Gender inequality affects various domains, including the economy [1], education [2], and health [3]. Female individuals have historically been disproportionately affected compared to their male counterparts [4]. In the field of medicine, gender inequities manifest in several ways, including differences in salaries, leadership roles, burnout rates, and workplace discrimination [5].

In recent years, there has been a growing participation of female individuals in the medical profession [6], influencing various aspects of healthcare, including the doctor-patient relationship and the delivery of health services [7]. Research indicates that females often exhibit enhanced communication skills with patients, are more likely to attend uninsured individuals, and show a preference for primary care specialties. Additionally, patients often prefer being treated by doctors who share their race and gender [8, 9].

In Peru, the participation of females in the health sector has significantly increased [10, 11]. For instance, the percentage of female physicians registered with the Medical College of Peru rose from 11.5% in 1971 to 48.9% by 2011 [12]. Similarly, within the Ministry of Health and regional governments, the percentage of female doctors increased from 34.6% in 2013 to 38.6% in 2018 [13].

The distribution of gender across various medical specialties remains underexplored and shows different trends over time. For instance, in Mexico, an increase of 3% in the participation of females in medical specialties and 9% in surgical specialties has been reported over a period of 14 years [14]. The proportion of female physicians in surgical specialties continues to be underrepresented, with a percentage ranging from 12% to 38.4% across most surgical specialties, with the exception of obstetrics and gynecology, with a percentage ranging from 58% to 82.7% [15–23]. These disparities may be attributed to factors, such as gender discrimination, lower job satisfaction among females, and higher levels of gender-based harassment [18, 24, 25]. Consequently, these barriers continue to obstruct the full integration of female individuals into medical specialties.

By 2024, no study has assessed gender disparities in the application and admission processes for various medical specialties in Peru. Therefore, the objective of this study was to evaluate the trends in female applications and admissions to different medical specialties in Peru between 2016 and 2023, and to determine whether being female is associated with a lower admission frequency.

## Methods

### Study design and data source

We conducted a cross-sectional study using panel data to assess the proportion of female applicants and admissions medical residency programs in Peru. Data were obtained from the public application and admission results provided by the "Consejo Nacional de Residentado Médico" (CONAREME) [26]. These results, which are publicly available, cover the application outcomes

for each year from 2016 to 2023. Data from years prior to 2016 were excluded from the study due to the unavailability of application results on the CONAREME website for those years.

## Variables

The gender variable was constructed based on applicants' names, as gender information was not explicitly provided in the application data. We utilized an existing database that predicts an individual's probable gender based on their first name [27], an approach employed in previous studies [28, 29]. In instances where gender could not be assigned from the first name, the second name was used following the same method. For instances in which gender could not be assigned using these names, two reviewers manually examined and assigned gender to names, adding 1,336 entries to the database of 46,445. Ultimately, 1,200 names (2.44% of the total) for which gender could not be determined using these methods were excluded from the analysis.

Additionally, for each applicant, the following variables were recorded: specialty of application, type of specialty (clinical specialties, including anesthesiology, family medicine, pediatrics, rheumatology, internal medicine, cardiology, physical medicine and rehabilitation, endocrinology, oncology, pulmonology, critical care medicine, hematology, radiation oncology, neurology, neonatology, nephrology, infectious diseases, geriatrics, psychiatry, gastroenterology, neurology, immunology, health administration, occupational medicine, sports medicine, anesthesiology, forensic medicine, interventional radiology; laboratory-diagnostic specialties, including medicine, radiology; medical-surgical specialties, including dermatology, obstetrics and gynecology, emergency medicine, ophthalmology, otolaryngology, traumatology and orthopedics, urology; and surgical specialties, including general surgery, surgical oncology, head and neck surgery, pediatric surgery, thoracic surgery, plastic surgery, hand surgery, ophthalmic surgery, neurosurgery), region of application (obtained from the university of application), and application result (admission or no admission).

For the trend analysis, the percentage of females among the applicants and those admitted to the medical residency program in Peru was considered, with these variables being of an ecological level. Finally, for the association analysis, the dependent variable was the admission outcome to the medical residency program, and the independent variable was the applicant's gender (female vs. male).

## Data analysis

Data handling and analysis were conducted using R programming language version 4.3.3. [30]. For descriptive analysis, we presented absolute and relative frequencies for all available records, as well as by each year of the study.

To examine changes in the percentage of female applicants and admissions to the medical residency program during the study period, we first calculated the percentage of female individuals among applicants and admissions by year and specialty. To ensure reliable estimates, we included only specialties with at least 100 applicants or admissions across the 8-year study period. These data were then used in multilevel linear regression models (LMM). The independent variable was the year of study (numerical, 2016 to 2023), and the outcome variables were: the percentage of female applicants (model 1) and the percentage of female admissions (model 2). We incorporated random intercepts and slopes by specialty (level 2 of the model) to account for variability in the percentage of female across specialties.

Although LMMs can use numerical optimization techniques, such as maximum likelihood estimation (MLE), Bayesian estimation frameworks have recently been adopted and show similar performance in identifying and characterizing trends and patterns over time [31, 32]. The decision to use Bayesian models was based on their ability to provide less biased estimates and

confidence intervals when dealing with scarce data, such as panel data. In our study, the data are grouped by year, resulting in 8 data points per specialty.

These models were therefore estimated using a Bayesian multilevel modeling framework using the brms package in R [33]. This approach provided a global intercept (representing the proportion of females in 2016) and a global beta coefficient (indicating the annual percentage change in female participation), accounting for variability between specialties. Additionally, the model estimates a different intercept and beta coefficient for each specialty, with results presented in graphical form to facilitate interpretation.

To evaluate whether being a woman was associated with admission, we used a multilevel Poisson regression model with robust variance. This model estimated the prevalence ratio (PR) and its corresponding 95% confidence interval (95% CI), accounting for variability between years and specialties. We compared the performance of the empty model with crude and adjusted models (considering region, year of application, and type of specialty) by presenting the Log-likelihood and Akaike information criterion values, with lower values indicating better model performance. This model was estimated using the lme4 package in R [34].

## Ethical considerations

This study did not require ethics committee approval because it involved analysis of publicly available secondary data. The data were obtained through the CONAREME website (https://www.conareme.org.pe/web/) [26]. Although the publicly available data included the names, surnames, and national identification numbers, these were removed after extracting the relevant variables for the final analysis.

## Results

From 2016 to 2023, a total of 49,213 individuals applied for medical residency programs. Gender was assigned to 48,013 applicants (97.56%). Of these, 23,065 (48%) were females and 24,948 (52%) were males. A year-on-year increase in the number of applicants was observed, except during 2020 and 2021.

## Characteristics of applicants to medical residency program

Table 1 summarizes the characteristics of the applicants according to the year of application, result, type of specialty, and region. The proportion of female applicants increased from 44.7% in 2016 to 50.6% in 2023. Of the total applicants, 38.9% were admitted to a medical residency program. Applicants were distributed across specialties as follows: 46.7% applied to medical specialties, 27.2% to medical-surgical specialties, 19.5% to surgical specialties, and 6.6% to laboratory-diagnostic specialties. Regionally, 72.5% of applicants were from Lima, 13.0% from the North, 10.3% from the South, 3.7% from the Center, and 0.4% from the East.

## Trends of female applicants in medical specialty

Using the Bayesian multilevel linear model, we found that the proportion of female applicants to the medical residency program in 2016 was, on average, 47.1%, with an average annual increase of 0.8% (95% CI: +0.4% to +1.3%) across specialties. This model provided insights into the percentage of female applicants as well as the trend over time within each specialty. In 2016, the specialties with the lowest proportion of female applicants were Trauma and Orthopedics (8.9%), Urology (16.1%), Cardiovascular and Thoracic Surgery (18.2%), and Neurosurgery (19.4%). In contrast, the specialties with the highest proportion of female applicants were Physical Medicine and Rehabilitation (71.9%), Dermatology (71.2%), Pathological Anatomy

**Table 1. Characteristics of applicants to the medical residency program in Peru between 2016 and 2023.**

| Characteristic | Total N = 48 013 | 2016 N = 5905 | 2017 N = 6258 | 2018 N = 6348 | 2019 N = 6503 | 2020 N = 4836 | 2021 N = 4966 | 2022 N = 6321 | 2023 N = 6876 |
|---|---|---|---|---|---|---|---|---|---|
| | n (%) | n (%) | n (%) | n (%) | n (%) | n (%) | n (%) | n (%) | n (%) |
| **Gender** | | | | | | | | | |
| Males | 24 948 (52.0) | 3263 (55.3) | 3404 (54.4) | 3398 (53.5) | 3341 (51.4) | 2389 (49.4) | 2532 (51.0) | 3222 (51.0) | 3399 (49.4) |
| Females | 23 065 (48.0) | 2642 (44.7) | 2854 (45.6) | 2950 (46.5) | 3162 (48.6) | 2447 (50.6) | 2434 (49.0) | 3099 (49.0) | 3477 (50.6) |
| **Application result** | | | | | | | | | |
| Admitted | 18 701 (38.9) | 2252 (38.1) | 2302 (36.8) | 2400 (37.8) | 2461 (37.8) | 2074 (42.9) | 2475 (49.8) | 2433 (38.5) | 2304 (33.5) |
| Not admitted | 29 312 (61.1) | 3653 (61.9) | 3956 (63.2) | 3948 (62.2) | 4042 (62.2) | 2762 (57.1) | 2491 (50.2) | 3888 (61.5) | 4572 (66.5) |
| **Specialty type**** | | | | | | | | | |
| Clinical | 22 409 (46.7) | 2796 (47.3) | 3022 (48.3) | 2995 (47.2) | 3134 (48.2) | 2406 (49.8) | 2401 (48.3) | 2742 (43.4) | 2913 (42.4) |
| Laboratory-diagnostics | 3145 (6.6) | 438 (7.4) | 477 (7.6) | 426 (6.7) | 413 (6.4) | 319 (6.6) | 346 (7.0) | 341 (5.4) | 385 (5.6) |
| Medical-surgical | 13 074 (27.2) | 1676 (28.4) | 1701 (27.2) | 1762 (27.8) | 1741 (26.8) | 1208 (25.0) | 1183 (23.8) | 1780 (28.2) | 2023 (29.4) |
| Surgical | 9385 (19.5) | 995 (16.9) | 1058 (16.9) | 1165 (18.4) | 1215 (18.7) | 903 (18.7) | 1036 (20.9) | 1458 (23.1) | 1555 (22.6) |
| **Region** | | | | | | | | | |
| Center | 1697 (3.7) | 181 (3.1) | 214 (3.4) | 251 (4.0) | 578 (8.9) | 138 (2.9) | 109 (2.7) | 103 (2.0) | 123 (1.8) |
| Lima | 33 283 (72.5) | 4385 (74.3) | 4581 (73.2) | 4698 (74.0) | 4417 (67.9) | 3758 (77.7) | 2841 (70.2) | 3584 (69.9) | 5019 (73.0) |
| North | 5973 (13.0) | 697 (11.8) | 782 (12.5) | 749 (11.8) | 827 (12.7) | 488 (10.1) | 572 (14.1) | 828 (16.2) | 1030 (15.0) |
| East | 198 (0.4) | 24 (0.4) | 33 (0.5) | 27 (0.4) | 32 (0.5) | 26 (0.5) | 29 (0.7) | 17 (0.3) | 10 (0.1) |
| South | 4749 (10.3) | 618 (10.5) | 648 (10.4) | 623 (9.8) | 649 (10.0) | 426 (8.8) | 498 (12.3) | 593 (11.6) | 694 (10.1) |

*The variable Region has 2113 missing values. 91 values in the year 2021 and 1196 values in the year 2022

**Clinical specialties, including anesthesiology, family medicine, pediatrics, rheumatology, internal medicine, cardiology, physical medicine and rehabilitation, endocrinology, oncology, pulmonology, critical care medicine, hematology, radiation oncology, neurology, neonatology, nephrology, infectious diseases, geriatrics, psychiatry, gastroenterology, neurology, immunology, health administration, occupational medicine, sports medicine, anesthesiology, forensic medicine, interventional radiology; laboratory-diagnostic specialties, including medicine, radiology; medical-surgical specialties, including dermatology, obstetrics and gynecology, emergency medicine, ophthalmology, otolaryngology, traumatology and orthopedics, urology; and surgical specialties, including general surgery, surgical oncology, head and neck surgery, pediatric surgery, thoracic surgery, plastic surgery, hand surgery, ophthalmic surgery, neurosurgery.

(71.1%), and Geriatrics (69.7%). An increasing trend in female applicants was observed in 36 specialties, except for Family Medicine (-0.6% annually), Hematology (-0.6% annually), Pediatric Neurology (-0.7% annually), and Pathological Anatomy (-0.9% annually), which showed a decreasing trend (**Fig 1**).

## Trends of female admissions to medical specialties

The proportion of females admitted to the residency program in 2016 was 47.1%, with an average annual increase of 0.9% (95% CI: +0.5% to +1.3%) across specialties. In 2016, the specialties with the lowest acceptance rates for females were Trauma and Orthopedics (8.4%), Urology (15.6%), Cardiovascular and Thoracic Surgery (17.7%), and Neurosurgery (18.9%). In contrast, specialties with the highest acceptance rates were Physical Medicine and Rehabilitation (71.9%), Dermatology (71.2%), Pathological Anatomy (71.2%), and Geriatrics (69.7%). An increasing trend in female acceptance was observed in 34 specialties, except for Family Medicine (-0.6% annually), Hematology (-0.6% annually), and Pathological Anatomy (-0.8% annually), which showed a decreasing trend (**Fig 2**).

## Association between gender and admission to medical residency programs

We used a multilevel Poisson regression model with robust variance, with specialties as random intercept and slope variables, to assess the association between gender and admission to a

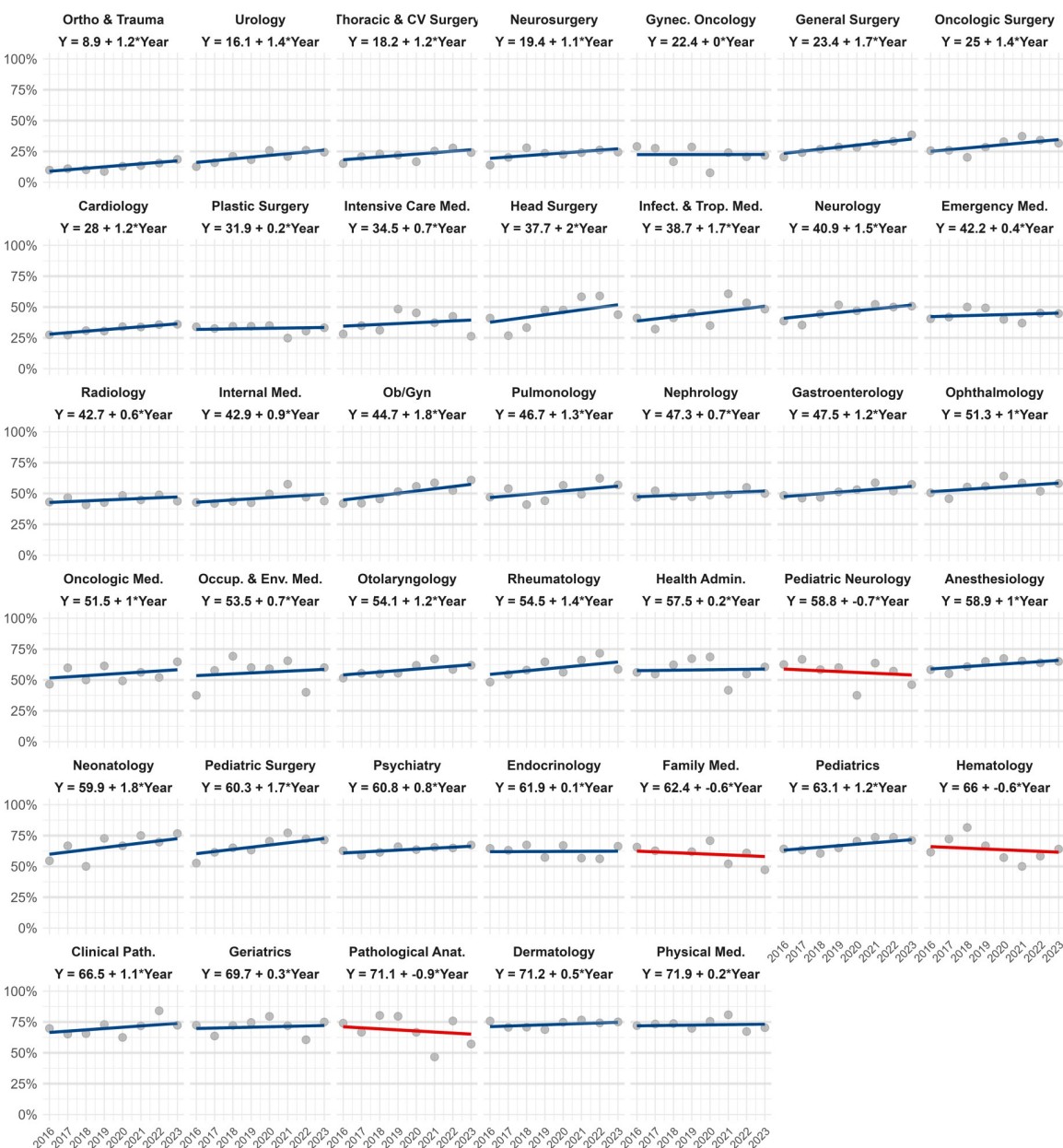

**Fig 1. Trends in female applicants to medical residency programs in Peru, 2016–2023.**

medical residency program. The analysis showed that female gender was associated with an 18% lower probability of admission to the medical residency program (PR: 0.82; 95% CI: 0.78–0.85; p-value: <0.001) compared to male. This result was statistically significant and independent of the region of application, year, and type of specialty (**Table 2**).

## Discussion

### Main findings

This study examined gender differences in the application and admission processes for medical residency programs in Peru between 2016 and 2023. In 2016, we found that the proportion

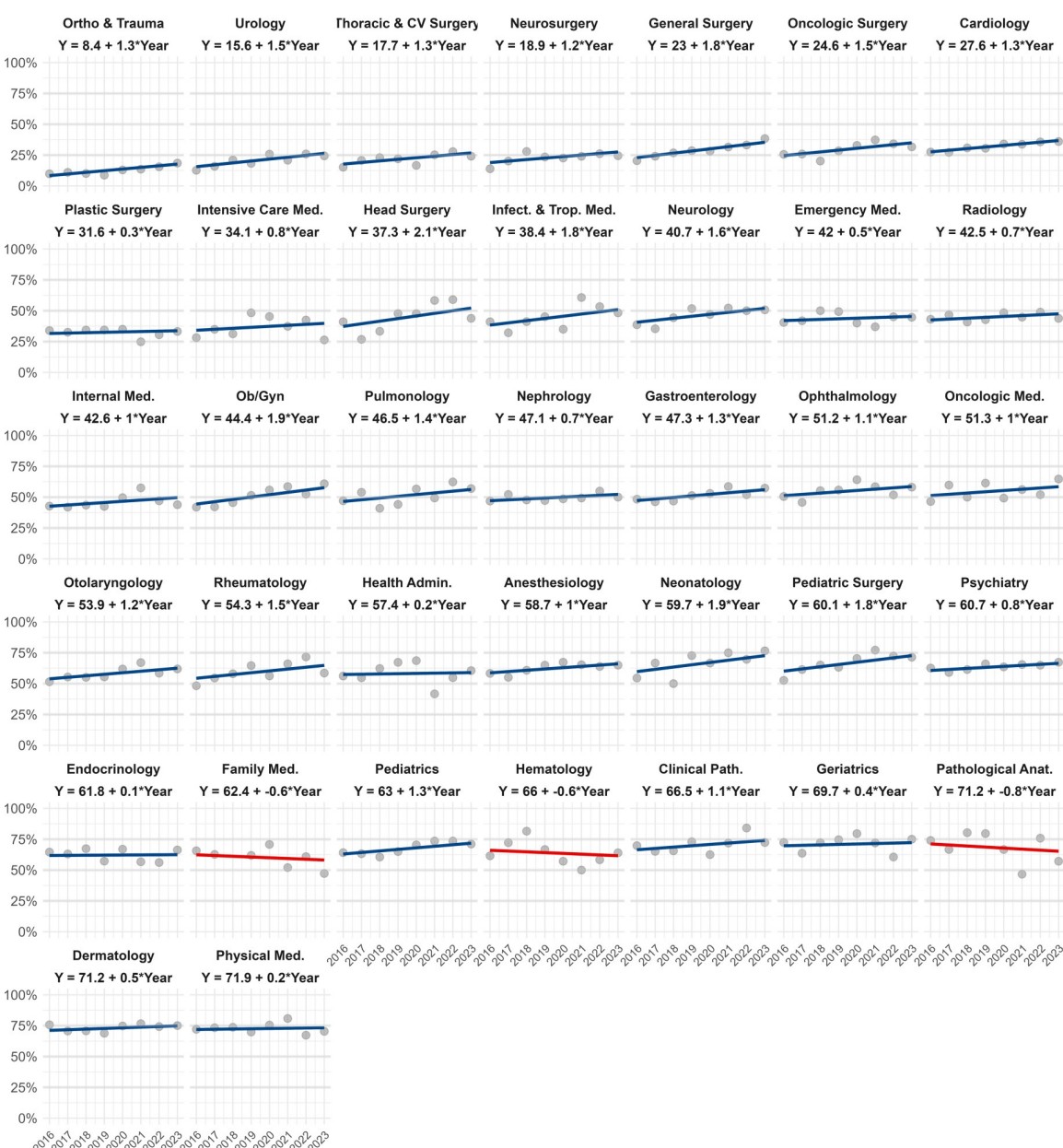

**Fig 2. Trends in female admissions to medical residency programs in Peru, 2016–2023.**

of female applicants was below 50% in 20 out of the 40 specialties evaluated. Additionally, 8 of the 10 specialties with the lowest proportion of female applicants were surgical specialties (Orthopedics and Traumatology, Urology, Thoracic and Cardiovascular Surgery, Neurosurgery, Gynecological Oncology, General Surgery, Oncological Surgery, and Plastic Surgery), while the 10 specialties with the highest proportion of female applicants were non-surgical (Physical Medicine and Rehabilitation, Dermatology, Pathology, Geriatrics, Clinical Pathology, Hematology, Pediatrics, Family Medicine, Endocrinology, and Psychiatry). Regarding the trend of female applicants, we observed an increase in 36 of the 40 specialties evaluated.

The increasing participation of women in medical specialties indicates progress toward greater gender equity. The Organization for Economic Co-operation and Development

**Table 2. Association between gender and admission to a medical residency program using multilevel Poisson regression models with robust variance.**

| Variables | Empty Model | | | Crude Model | | | Adjusted Model* | | |
|---|---|---|---|---|---|---|---|---|---|
| | cPR | 95% CI | p | cPR | 95% CI | P | aPR | 95% CI | p |
| **Gender** | | | | | | | | | |
| Male | Ref. | | | Ref. | | | Ref. | | |
| Female | - | - | - | 0.83 | 0.79–0.86 | <0.001 | 0.82 | 0.78–0.85 | <0.001 |
| **ICC** | 0.22 | | | 0.20 | | | 0.11 | | |
| **Log-likelihood** | -30496.51 | | | -30453.57 | | | -28900.66 | | |
| **AIC** | 60997.02 | | | 60913.14 | | | 57835.31 | | |

cPR: Crude prevalence ratio; aPR: Adjusted prevalence ratio; CI: confidence interval, ICC: Interclass correlation coefficient; AIC: Akaike information criterion

* Model adjusted for region, application year, and type of specialty; random intercept and slopes by specialty were considered.

(OECD) has reported an increase in the proportion of women participating in the medical profession across various countries [35]. This trend may be attributed to factors such as increased female role models, changes in social perceptions of women's roles in medicine and policies promoting equal opportunities [36, 37]. In Peru, the Law on Equal Opportunities between Women and Men (Law No. 28983) ensures equality in rights, opportunities, and resources across areas like education, economy, and politics [38]. Furthermore, the National Policy on Gender Equality (PNIG), led by the Ministry of Women and Vulnerable Populations, sets key goals to reduce gender disparities across various sectors [39]. Additionally, there has been an increase in the presence of females in medical schools, which translates to a higher future presence in various specialties and the medical workforce [37, 40]. However, there is still an imbalance in the participation of females across different specialties. This aligns with a previous study showing that the percentage of female residents in surgical and non-surgical specialties in the United States from 2010 to 2019 is below 45% [41]. These findings highlight the need for an inclusive environment the encourages females participation in all areas of medicine, particularly in those in which they are underrepresented.

It is important to note that the specialties to which males and females most frequently apply differ. For example, males show a greater preference for surgical specialties [42, 43]. These results are consistent with both national [13] and international data [44–46] showing male predominance in these fields. This may be explained by factors such as the perception of a less inclusive work environment, lack of female role models, and the physical and time demands associated with these specialties [41]. On the other hand, non-surgical specialties show a higher proportion of female applicants, suggesting that these fields may be perceived as more accessible and compatible with female applicants' personal and professional preferences [41]. Addressing gender imbalances in surgical residency programs may require interventions like mentorship programs that provide support and networking opportunities for female residents. Additionally, adopting gender equity policies, such as extended maternity leaves and impartial selection processes, can help eliminate barriers faced disproportionately by females [47]. Finally, making structural changes within the residency program, such as offering flexible schedules and robust institutional support, is crucial to facilitating work-life balance, particularly in specialties with low female representation [48].

Regarding individuals who were admitted, we found similar results correlating with the proportion of applicants. The proportion of females admitted in 2016 was below 50% in 19 of the 37 specialties evaluated. Additionally, 8 of the 10 specialties with the fewest females applicants admitted were surgical specialties (Orthopedics and Traumatology, Urology, Thoracic

and Cardiovascular Surgery, Neurosurgery, General Surgery, Oncological Surgery, Plastic Surgery, and Head and Neck Surgery), while the 10 specialties with the highest proportion of females applicants admitted were non-surgical specialties (Physical Medicine and Rehabilitation, Dermatology, Pathology, Geriatrics, Clinical Pathology, Hematology, Pediatrics, Family Medicine, Endocrinology, and Psychiatry). We observed an increase in the number of females admitted in 34 of the 37 specialties evaluated.

Finally, we found that females are 18% less likely to enter the medical residency program compared to their male counterparts. Previous studies evaluating the differences between the number of male and female medical school graduates and those who are eventually admitted to a specialty suggest that these differences may be due to biases in the selection of applicants for residency programs [41]. A study examining recommendation letters for ophthalmology residency applicants [49] determined that letters written for male applicants used more "authentic" words, contained more "leisure" words and fewer "feel" words and "biological processes" than those written for female applicants. The letters of female applicants also had fewer adjectives describing skills such as "analytical" or "genius" [49]. However, these studies were conducted in the United States, where the selection of participants involves significant subjective components, such as recommendation letters, analysis of scientific publications, and clinical and work experience [42, 50, 51]. In Peru, the selection process does not include these components, and the total score is obtained numerically from: the residency exam score, the rural service score (SERUMS), the National Medical Examination (ENAM) score, undergraduate grade point average, being in the top 20% of the undergraduate class, and additional points for years of service to the country [52]. Therefore, since the process is numerical, the observed discrepancies cannot be easily explained by the same subjective biases as in the United States. Studies focused on differences in academic performance between males and females in exams and academic performance have yielded inconclusive results, with some studies indicating better academic performance among males, while others indicate the opposite [53–56]. Further research is needed to understand the discrepancies in residency program application outcomes between males and females in Peru.

Another significant finding was the year-on-year increase in the number of applicants, with a temporary decrease in 2020 and 2021, likely due to the COVID-19 pandemic. The pandemic reduced medical residency applicants in Peru due to several factors. Financial challenges and economic instability, the interruptions in medical education from suspension of clinical training [57] and the health risks associated with COVID-19, both for potentials applicants and their families, created emotional and physical stress [58] that influenced some to defer their medical specialization plans.

## Implications of the study

This study, conducted in Peru, a developing country, highlights important implications for gender equity in specialized medical training. In a context in which healthcare infrastructure and professional opportunities may be limited, our findings underscore the need for policies that promote greater inclusion and representation of females in most medical specialties.

Our findings indicate that, despite the increasing participation of females in the application process across many specialties, they are still less likely to be admitted to residency programs. Factors such as lower exposure to certain specialties during training and personal preferences shaped by social and professional environments may contribute to this disparity [59, 60].

Surgical specialties, such as Traumatology, Orthopedics, Urology, Cardiovascular Surgery, and Neurosurgery, where females are underrepresented, require particular attention to understand and reduce barriers to female participation [36, 37, 41]. In contrast, specialties like

Physical Medicine and Rehabilitation, Dermatology, and Geriatrics have a higher proportion of female applicants and admitted females, suggesting a need to study what factors contribute to this positive disparity and how they can be replicated in other areas.

Finally, the findings of this study should prompt policymakers and educational institutions to implement strategies that promote gender equality in medical education. This will not only benefit females but will also improve the overall quality of medical care, as previous studies have shown that including female physicians can positively impact patient health outcomes [9].

## Limitations and strengths

We acknowledge that gender was indirectly determined based on names. However, we believe that using names provides a reasonable approximation, with only 2.4% of respondents remaining undetermined using this method. Second, while we describe that application and admission rates were lower for females, we did not determine the causes of these differences. Understanding the trends observed still requires evaluation, possibly including qualitative studies. Third, our findings are limited by the use of binary gender categories, as we lack information on the gender identity of each applicant. The collection of this information is not included in the options for this non-anonymized database. This may limit the generalizability of the results to populations with diverse gender identities. The implications of excluding non-binary gender from our analysis may result in alterations to the observed trends for females in certain medical specialties. Nevertheless, the estimated prevalence of potential non-binary applicants to medical residency programs is likely to be less than 5% [61], suggesting that any variations in gender frequency may not be substantial. Fourth, we recognize that we could not assess trends in applications or admissions for specialties with a low number of applicants (fewer than 100 applicants during the study period) due to potential bias from the small sample size each year. Fifth, we preferred to use a dichotomous variable instead of average scores to evaluate gender disparities in admissions, as this choice facilitates a clearer and more direct interpretation of the results, avoiding the masking of variations across different specialties. Additionally, scores may be influenced by the difficulty of the exam and the academic context, complicating comparisons across years. However, we recognize that an analysis of scores could provide additional insights into the relationship between academic performance and admission rates, which could be addressed in future research. Finally, trends could vary with more years of follow-up, making it worthwhile to update this study. The lack of data on admission outcomes from previous years also limits our understanding of temporal trends in applications and admissions.

Despite these limitations, this is the first study to describe gender differences and discrepancies in application outcomes between males and females in Peru. Additionally, we used a multilevel analytical approach to model trends in female applicants and entrants, considering variability between specialties each year.

## Conclusions

This study reveals a persistent gender disparity in the application and admission processes for medical residency programs in Peru from 2016 to 2023. While the proportion of female applicants has generally increased, this trend is not consistent across all specialties. Specialties such as Family Medicine, Hematology, Pediatric Neurology, and Pathology have shown either stagnation or a decline in the percentage of female applicants. Furthermore, the study reveals that females are 18% less likely to be admitted to residency programs compared to their male counterparts. Specialties with the lowest proportion of female applicants and admissions include Trauma and Orthopedics, Urology, Cardiovascular and Thoracic Surgery, and Neurosurgery.

These findings highlight the need for targeted interventions to address gender disparities, particularly in underrepresented specialties. Such actions are important to fostering equitable opportunities and enhancing the diversity and quality of the medical workforce.

## Acknowledgments

The authors thank Universidad Científica del Sur for covering the APC and for the English editing support provided by Donna Pringle.

## Author Contributions

**Conceptualization:** Daniel Medina-Neira, Daniel Fernandez-Guzman.

**Data curation:** Daniel Medina-Neira.

**Formal analysis:** Daniel Medina-Neira, Daniel Fernandez-Guzman.

**Investigation:** Daniel Medina-Neira, Brenda Caira-Chuquineyra.

**Methodology:** Daniel Medina-Neira, Brenda Caira-Chuquineyra, Daniel Fernandez-Guzman.

**Supervision:** Daniel Medina-Neira, Brenda Caira-Chuquineyra, Daniel Fernandez-Guzman.

**Visualization:** Daniel Fernandez-Guzman.

**Writing – original draft:** Daniel Medina-Neira, Brenda Caira-Chuquineyra, Daniel Fernandez-Guzman.

**Writing – review & editing:** Daniel Medina-Neira, Brenda Caira-Chuquineyra, Daniel Fernandez-Guzman.

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
