## [Decision Letter · Decision Letter 0]

11 Oct 2024

PONE-D-24-38457Gender disparities in the medical residency program in Peru: an analysis of applications and admissions from 2016 to 2023.PLOS ONE

Dear Dr. Fernández Guzmán,

Thank you for submitting your manuscript to PLOS ONE. After careful consideration, we feel that it has merit but does not fully meet PLOS ONE’s publication criteria as it currently stands. Therefore, we invite you to submit a revised version of the manuscript that addresses the points raised during the review process.

We look forward to receiving your revised manuscript.

Kind regards,

Barry Kweh

Academic Editor

PLOS ONE

**Journal Requirements:**

**Additional Editor Comments:**

A well written article but a broader discussion of the limitations and potential biases inherent to specialty selection is required. A broader discussion of the comparative literature regarding gender disparities in other countries would also be valuable.

Reviewers' comments:

Reviewer's Responses to Questions

**Comments to the Author**

1. Is the manuscript technically sound, and do the data support the conclusions?

Reviewer #1: Yes

Reviewer #2: Yes

Reviewer #3: Partly

Reviewer #4: Partly

2. Has the statistical analysis been performed appropriately and rigorously? 

Reviewer #1: Yes

Reviewer #2: I Don't Know

Reviewer #3: Yes

Reviewer #4: N/A

3. Have the authors made all data underlying the findings in their manuscript fully available?

Reviewer #1: Yes

Reviewer #2: No

Reviewer #3: Yes

Reviewer #4: Yes

4. Is the manuscript presented in an intelligible fashion and written in standard English?

Reviewer #1: No

Reviewer #2: Yes

Reviewer #3: Yes

Reviewer #4: Yes

5. Review Comments to the Author

**Reviewer #1:** Comments by Headings

1. General comment

The authors could have used line numbers and page numbers to ease the review process. In its current form, it is difficult to point authors to specific locations for suggested corrections.

2. Abstract

The abstract provides a clear overview of the study, highlighting the main findings effectively. The authors have succeeded in presenting the relevance of their research to gender disparities in medical residency applications in Peru, which is a commendable approach to addressing an underexplored topic.

Suggestions for Improvement:

To enhance readability, authors may consider rearranging the results to present proportions first, followed by the trend analysis, and then the measures of association. This sequence will provide a clearer flow from descriptive to inferential statistics, making it easier for readers to follow the findings.

3. Introduction

The introduction does an excellent job of setting the stage by providing a thorough background on gender disparities in medical specialties. The objectives are clearly defined, and the importance of the study is well articulated. The inclusion of data from various countries to support the need for gender equity in medical specialties is particularly strong.

Suggestions for Improvement:

a. Clarity in Terminology: The opening sentence, "differences between individuals," could be more specific in its reference to gender-related disparities. I suggest explicitly stating that these differences pertain to unequal treatment or outcomes based on gender, to avoid ambiguity.

b. References: The following statements should be supported with references:

"The percentage of female physicians registered with the medical college of Peru rose from 11.5% in 1971 to 48.9% by 2011."

"In the Ministry of Health and regional governments, the percentage of female doctors rose from 34.6% in 2013 to 38.6% in 2018."

c. Precision in Time Statements: The expression “To date” in the last paragraph needs to specify the year, as it currently lacks clarity. For example, “As of [year],” will enhance temporal specificity.

4. Methods

The authors’ choice of using a multilevel model to assess gender disparities across medical specialties is a robust approach. It accounts for variations between specialties and time periods, which adds rigor to the analysis.

Suggestions for Improvement:

a. Gender Classification: The binary classification of gender (male and female) does not account for non-cisgender individuals, which could potentially skew the results. I suggest the authors acknowledge this limitation and discuss how the inclusion of non-binary or transgender categories could affect the study’s findings.

b. Outcome and Independent Variables: The structure of the methods section could benefit from a clearer organization. I recommend introducing the outcome and independent variables explicitly under a separate subsection before diving into detailed explanations.

5. Results

The results section presents a comprehensive analysis of trends in female participation in medical residency programs. The use of percentages and graphs enhances understanding of gender disparities, and the breakdown by specialty provides valuable insights.

Suggestions for Improvement:

a. Time Period Wording: The phrase "Between 2016 and 2023" should be revised to "From 2016 to 2023" for clarity and accuracy, as it better reflects the inclusion of both years in the data.

b. Relocation of Explanations: The explanation for the observed application rate drop in 2020 and 2021 could be better placed in the discussion section, where the impact of external factors like the COVID-19 pandemic can be explored in depth.

c. Table 1 Descriptions: To improve clarity, I recommend either providing a descriptive summary of each specialty in Table 1. If the authors intend to use groupings (medical, laboratory-diagnostic, medical-surgical) as they have, they should provide a key below the table to explain what the groupings represent.

d. Subheadings: The inclusion of subheadings in the results section would greatly enhance the clarity and organization of the section, especially considering the detailed nature of the findings.

e. Graph Titles: For the graphs, I recommend that the authors review the PLOS ONE submission guidelines on presenting graph titles and improve the graph titles. This will help readers quickly grasp the key points/legends when looking at the figures.

f. Consistency in Statistical Models: Consistency in Statistical Models: The model described in the methods section for measuring the association is multilevel Poisson regression, while the model in the results section is a generalized linear mixed-effects model. I recommend that the authors use consistent terminology to avoid confusion for the reader.

6. Discussion

The discussion does well to contextualize the findings within the broader literature and highlight gender disparities in medical specialties, particularly in surgical fields. The comparison with international studies strengthens the relevance of the findings.

Suggestions for Improvement:

a. When discussing the proportion of women admitted to specialties, it would be beneficial to expand beyond the year 2016. Since data from 2016–2023 is available, a broader temporal analysis will provide a more comprehensive understanding of trends and fluctuations over time.

b. Addressing Gender Imbalance: The authors could provide a more detailed exploration of potential solutions to gender imbalances, particularly in surgical specialties. This would elevate the discussion by not only identifying disparities but also suggesting actionable interventions, such as mentorship programs, policies, or structural changes in medical training environments.

c. Limitation on Gender Categorization: While the discussion acknowledges limitations, the binary gender classification should be discussed more critically. Addressing how this might exclude or marginalize non-binary individuals would show a broader inclusivity in the analysis.

Conclusion

The paper addresses an important gap in the literature on gender disparities in medical residency programs in Peru and offers valuable insights that will contribute to the field. With some refinement in the clarity of methods, results, and expansion in the discussion, this manuscript will make a strong contribution to the conversation on gender equity in medicine.

**Reviewer #2:** Results - Page 13: Perhaps elaborate on what specifically during the pandemic in 2020 to 2021 prevented the increase in applicants (financial costs, pause in schooling, illness?).

Discussion - Page 18: Perhaps add a sentence elaborating on the "institutional and governmental policies promoting equal opportunities" in Peru, as those from outside the country may not be aware of the policies.

Figures - May be nice to include a bar graph comparing male and female admission of all specialties for a quick visual representation of the results.

References - May be good to double check reference style as some journal names are abbreviated (36, 40, 41) while others aren't (33, 43) etc.

**Reviewer #3:** This study addresses a significant issue (Gender disparities in the medical residency program in Peru), providing a detailed analysis of trends over time and across medical specialties. The study has an explanatory introduction, a review of relevant literature, and a thorough explanation of the methodology.

In the discussion part, the authors acknowledge that inferring the applicants gender based on their first names, using a binary gender classification, and excluding 2.4% of applicants limit the generalizability of their findings.

However, there are areas where the manuscript could be improved:

1- The study states that the application process is based on a numerical scoring system, in order to conclude more precisely that there is a gender disparity in the admissions, there should be a reference to the average score of each gender group ( or referring to its impact in the limitation part), and an explanation on the possible impact of gender on this scoring system.

2- The study mentions that women tend to apply more to certain specialties, where admission rates may be more competitive. A disparity in specialty selection could skew the overall admission rates for women if certain specialties are more difficult to enter. It is essential to address the total application and admission rate by specialty to provide a more accurate picture.

3- The study mentions that some data were excluded. However, it’s unclear if the authors conducted a sensitivity analysis to check whether the exclusion of this data affects the overall results.

**Reviewer #4:** While the manuscript is sound, the data can only partially support the conclusions. as stated in the research limitations, the sex of participants was made based on inferences from the name which by default lowers the accuracy. As such it will affect the significance of the study's findings. In addition, the title of the research does not elaborate well on what has been done and I recommend including the type of study applied in the title. on the abstract the first sentence of methods seems like a repetition of the objective written in different words. Instead, I recommend describing the type of study design used as an introduction to the method just like what is written in the methods part on the full article. on the introduction, you have used outdated references that were dated more than 15 years back such as reference no. 9, I recommend removing them and finding new references. In addition, little information given on global perspective distribution of gender across various medical specialties. it will be better to add more research papers.

when it comes to methods part, I believe the source of the data used for the study is inadequate. I would recommend getting the names of the applicants and correlating it with the universities record for more detailed information such as sex of participants. The other thing is I do not believe the study has exhausted in researching independent variables which may affect the result of study. The only independent variable included in the study is the year of study while variables such as financial status or Admission Requirement status of the participants have not been touched. such confounding variables must first be addressed before concluding that there is gender disparity in the admission for medical residency program. on the result section on the sociodemographic data instead of putting a single year's data on the table I recommend using range or overall average to include other years emphasizing on years with highest female. the range or overall average result should also be used in abstract so that the abstract information will not be misleading. In addition, model assumption or model adequacy tests done for regression have not been mentioned, this should be included as part of the result.

Overall the research has a good start but as stated in the limitations needs more work on it to reach into an accurate conclusion

6. PLOS authors have the option to publish the peer review history of their article (what does this mean?). If published, this will include your full peer review and any attached files.

Reviewer #1: No

Reviewer #2: No

Reviewer #3: No

Reviewer #4: **Yes: **Dr. Anteneh A. Beyene

---

## [Author Response · Author response to Decision Letter 0]

11 Nov 2024

"See the response to comments in the file 'Response to Reviewers'."

Reviewer Comments Response

#1 The authors could have used line numbers and page numbers to ease the review process. In its current form, it is difficult to point authors to specific locations for suggested corrections. 

We appreciate your comment.

We have added line numbers and page numbers.

#1 The abstract provides a clear overview of the study, highlighting the main findings effectively. The authors have succeeded in presenting the relevance of their research to gender disparities in medical residency applications in Peru, which is a commendable approach to addressing an underexplored topic.

Suggestions for Improvement:

To enhance readability, authors may consider rearranging the results to present proportions first, followed by the trend analysis, and then the measures of association. This sequence will provide a clearer flow from descriptive to inferential statistics, making it easier for readers to follow the findings. 

We appreciate your comment.

The following order of results presentation has been followed in the summary: descriptive and trend results of applicants, followed by those of admitted students. 

#1 The introduction does an excellent job of setting the stage by providing a thorough background on gender disparities in medical specialties. The objectives are clearly defined, and the importance of the study is well articulated. The inclusion of data from various countries to support the need for gender equity in medical specialties is particularly strong.

Suggestions for Improvement:

a. Clarity in Terminology: The opening sentence, "differences between individuals," could be more specific in its reference to gender-related disparities. I suggest explicitly stating that these differences pertain to unequal treatment or outcomes based on gender, to avoid ambiguity.

b. References: The following statements should be supported with references:

"The percentage of female physicians registered with the medical college of Peru rose from 11.5% in 1971 to 48.9% by 2011."

"In the Ministry of Health and regional governments, the percentage of female doctors rose from 34.6% in 2013 to 38.6% in 2018."

c. Precision in Time Statements: The expression “To date” in the last paragraph needs to specify the year, as it currently lacks clarity. For example, “As of [year],” will enhance temporal specificity 

We appreciate your comment.

a. We have modified the opening sentence to be more specific about gender-related differences. We modified the "Introduction" section in lines 62-63 to clarify what was done: “Disparities among individuals due to gender inequality across various domains such as the economy [1], education [2], and health [3]”.

b. We supported the mentioned statements with respective references. We modified the "Introduction" section in lines 75-76, by adding the references. 

c. We replaced it with a more specific time expression. We modified the "Introduction" section in line 84 to clarify what was done: “Up to 2024, no study has assessed gender disparities in the application and admission processes for various medical specialties in Peru”

#1 The authors’ choice of using a multilevel model to assess gender disparities across medical specialties is a robust approach. It accounts for variations between specialties and time periods, which adds rigor to the analysis.

Suggestions for Improvement:

a. Gender Classification: The binary classification of gender (male and female) does not account for non-cisgender individuals, which could potentially skew the results. I suggest the authors acknowledge this limitation and discuss how the inclusion of non-binary or transgender categories could affect the study’s findings.

b. Outcome and Independent Variables: The structure of the methods section could benefit from a clearer organization. I recommend introducing the outcome and independent variables explicitly under a separate subsection before diving into detailed explanations. 

We appreciate your comment.

a. We added this limitation in the corresponding section of the article and we discuss how inclusion of non-binary categories could affect the study results. We modified the " Limitations and Strengths" section in lines 319-326 to clarify what was done: “Third, our findings are limited by the use of binary gender categories, as we lack information on the gender identity of each applicant. The collection of this information is not included in the options for this non-anonymized database. This may limit the generalizability of the results to populations with diverse gender identities. The implications of excluding non-binary gender from our analysis may result in alterations to the observed trends for women in certain medical specialties. Nevertheless, the estimated prevalence of potential non-binary applicants to medical residency programs is likely to be less than 5%, suggesting that any variations in gender frequency may not be substantial”. 

b. A paragraph was added in the Methods section clarifying the variables used for trend analysis. Additionally, the outcome and independent variables used for the association analysis were specified. We modified the "Variables" section in lines 97-100, to clarify what was done: “We constructed the gender variable, based on applicants' names recorded in the database for the medical residency program in Peru, since gender information was not explicitly provided in the application process” and in lines 110-114, to clarify what was done: “For the trend analysis, the percentage of females among the applicants and those admitted to the medical residency program in Peru was considered, with these variables being of an ecological level. Finally, for the association analysis, the dependent variable was the admission outcome to the medical residency program, and the independent variable was the applicant's gender (female vs. male).”

#1 The results section presents a comprehensive analysis of trends in female participation in medical residency programs. The use of percentages and graphs enhances understanding of gender disparities, and the breakdown by specialty provides valuable insights.

Suggestions for Improvement:

a. Time Period Wording: The phrase "Between 2016 and 2023" should be revised to "From 2016 to 2023" for clarity and accuracy, as it better reflects the inclusion of both years in the data.

b. Relocation of Explanations: The explanation for the observed application rate drop in 2020 and 2021 could be better placed in the discussion section, where the impact of external factors like the COVID-19 pandemic can be explored in depth.

c. Table 1 Descriptions: To improve clarity, I recommend either providing a descriptive summary of each specialty in Table 1. If the authors intend to use groupings (medical, laboratory-diagnostic, medical-surgical) as they have, they should provide a key below the table to explain what the groupings represent.

d. Subheadings: The inclusion of subheadings in the results section would greatly enhance the clarity and organization of the section, especially considering the detailed nature of the findings.

e. Graph Titles: For the graphs, I recommend that the authors review the PLOS ONE submission guidelines on presenting graph titles and improve the graph titles. This will help readers quickly grasp the key points/legends when looking at the figures.

f. Consistency in Statistical Models: Consistency in Statistical Models: The model described in the methods section for measuring the association is multilevel Poisson regression, while the model in the results section is a generalized linear mixed-effects model. I recommend that the authors use consistent terminology to avoid confusion for the reader. 

We appreciate your comment.

a. We replaced the time period wording. We modified the "Results" section in line 153-154 to clarify what was done: “From 2016 to 2023, a total of 49,213 individuals applied for medical residency programs”

b. We added an explanation for the application rate drop during the COVID-19 pandemics and we placed it in the discussion section. We modified the "Discussion" section in lines 289-294 to clarify what was done: “Another significant finding was the year-on-year increase in the number of applicants, with the exception of 2020 and 2021, likely due to the COVID-19 pandemic. The pandemic reduced medical residency applicants in Peru due to several factors. Financial challenges and economic instability, the interruptions in medical education from suspension of clinical training and the health risks associated with COVID-19, both for potentials applicants and their families, created emotional and physical stress that influenced some to defer their medical specialization plans.”

c. We provide a key below the table to provide a descriptive summary of each specialty. We modified the "Results" section in line 170 to clarify what was done: “”. 

e. We have corrected the label and title size according to the journal suggestions. We modified the "Results" section in line 184-185 and 197-198 to clarify what was done: “Fig 1. Trends in Female Applicants to Medical Residency Programs in Peru, 2016–2023” and “Fig 2. Trends in Female Admissions to Medical Residency Programs in Peru, 2016–2023”. 

f. We used a consistent terminology to describe the used methods for measuring the association. We modified the "Results" section in line 200 to clarify what was done: “We used a multilevel Poisson regression model, with specialties as random intercept and slope variables, to assess the association between sex and admission to a medical residency program.”, and in lines 207-208 to clarify what was done: “Table 2: Association between gender and admission to a medical residency program using multilevel Poisson regression models”

#1 The discussion does well to contextualize the findings within the broader literature and highlight gender disparities in medical specialties, particularly in surgical fields. The comparison with international studies strengthens the relevance of the findings.

Suggestions for Improvement:

a. When discussing the proportion of women admitted to specialties, it would be beneficial to expand beyond the year 2016. Since data from 2016–2023 is available, a broader temporal analysis will provide a more comprehensive understanding of trends and fluctuations over time.

b. Addressing Gender Imbalance: The authors could provide a more detailed exploration of potential solutions to gender imbalances, particularly in surgical specialties. This would elevate the discussion by not only identifying disparities but also suggesting actionable interventions, such as mentorship programs, policies, or structural changes in medical training environments.

c. Limitation on Gender Categorization: While the discussion acknowledges limitations, the binary gender classification should be discussed more critically. Addressing how this might exclude or marginalize non-binary individuals would show a broader inclusivity in the analysis.

Conclusion

The paper addresses an important gap in the literature on gender disparities in medical residency programs in Peru and offers valuable insights that will contribute to the field. With some refinement in the clarity of methods, results, and expansion in the discussion, this manuscript will make a strong contribution to the conversation on gender equity in medicine. 

We appreciate your comment.

a. No data on the results of those admitted to the medical residency program in Peru is available for years prior to the 2016–2023 period. Consequently, data from applicants before 2016 were not included in our study’s analysis. This limitation is noted in the Strengths and Limitations section of the Discussion. 

b. We have expanded the discussion to address the gender imbalance in applicants and those admitted to surgical specialties with the respective suggestions for implementing interventions. We modified the "Discussion" section in lines 252-258 to clarify what was done: “To address the gender imbalance in surgical residency programs, it is essential to implement interventions such as mentorship programs that provide support and networking opportunities for female residents. Additionally, adopting gender equity policies, such as extended maternity leaves and impartial selection processes, can help eliminate barriers faced disproportionately by women. Finally, making structural changes within the residency program, such as offering flexible schedules and robust institutional support, is crucial to facilitating work-life balance, particularly in specialties with low female representation”.

c. We have expanded on the implications of excluding non-binary individuals in the limitations section of the discussion. We modified the "Limitations and Strengths" section in lines 319-326 to clarify what was done: “Third, our findings are limited by the use of binary gender categories, as we lack information on the gender identity of each applicant. The collection of this information is not included in the options for this non-anonymized database. This may limit the generalizability of the results to populations with diverse gender identities. The implications of excluding non-binary gender from our analysis may result in alterations to the observed trends for women in certain medical specialties. Nevertheless, the estimated prevalence of potential non-binary applicants to medical residency programs is likely to be less than 5%, suggesting that any variations in gender frequency may not be substantial”.

Reviewer Comments Response

#2 Results - Page 13: Perhaps elaborate on what specifically during the pandemic in 2020 to 2021 prevented the increase in applicants (financial costs, pause in schooling, illness?). 

We appreciate your comment.

We add an explanation on what prevented the increase in applicants during the pandemic in the discussion section. We modified the "Discussion" section in lines 289-294 to clarify what was done: “Another significant finding was the year-on-year increase in the number of applicants, with the exception of 2020 and 2021, likely due to the COVID-19 pandemic. The pandemic reduced medical residency applicants in Peru due to several factors. Financial challenges and economic instability, the interruptions in medical education from suspension of clinical training and the health risks associated with COVID-19, both for potentials applicants and their families, created emotional and physical stress that influenced some to defer their medical specialization plans.”

#2 Discussion - Page 18: Perhaps add a sentence elaborating on the "institutional and governmental policies promoting equal opportunities" in Peru, as those from outside the country may not be aware of the policies. 

We appreciate your comment.

We add a sentence explaining those institutional and governmental policies in Peru in the discussion section. We modified the "Discussion" section in lines 229-236 to clarify what was done: “This phenomenon can be attributed to several factors, including increased availability of female role models in various specialties, changes in social perceptions of women's roles in medicine and institutional and governmental policies promoting equal opportunities [33,34]. In Peru, the Law on Equal Opportunities between Women and Men (Law No. 28983) ensures equality in rights, opportunities, and resources across areas like education, economy, and politics. Furthermore, the National Policy on Gender Equality (PNIG), led by the Ministry of Women and Vulnerable Populations, sets key goals to reduce gender disparities across various sectors”. 

#2 Figures - May be nice to include a bar graph comparing male and female admission of all specialties for a quick visual representation of the results. 

We appreciate your comment.

We believe that the trend graph displaying the proportion of females in each specialty enables us to infer the proportion of males, as gender is represented as a dichotomous variable in our study. 

#2 References - May be good to double check reference style as some journal names are abbreviated (36, 40, 41) while others aren't (33, 43) etc. 

We appreciate your comment.

We standar

---

## [Editor Report · Decision Letter 1]

29 Nov 2024

PONE-D-24-38457R1Gender disparities in application and admission to the medical residency program in Peru: a cross sectional study from 2016 to 2023PLOS ONE

Dear Dr. Fernández Guzmán,

Thank you for submitting your manuscript to PLOS ONE. After careful consideration, we feel that it has merit but does not fully meet PLOS ONE’s publication criteria as it currently stands. Therefore, we invite you to submit a revised version of the manuscript that addresses the points raised during the review process.

We look forward to receiving your revised manuscript.

Kind regards,

Barry Kweh

Academic Editor

PLOS ONE

**Journal Requirements:**

**Additional Editor Comments:**

The authors have satisfactorily restructured their manuscript, broadened their discussion and clarified methodological strategy. Some proof reading and grammatical correction would significantly improve the quality and readability of the manuscript.

---

## [Author Response · Author response to Decision Letter 1]

11 Dec 2024

Attached are the answers in the document “answers to reviewers”.

---

## [Editor Report · Decision Letter 2]

17 Dec 2024

Gender disparities in application and admission to the medical residency program in Peru: a cross sectional study from 2016 to 2023

PONE-D-24-38457R2

Dear Dr. Guzman,

We’re pleased to inform you that your manuscript has been judged scientifically suitable for publication and will be formally accepted for publication once it meets all outstanding technical requirements.

Kind regards,

Barry Kweh

Academic Editor

PLOS ONE

Additional Editor Comments (optional):

The authors have broadened their discussion, clarified their methodology and improved their consistency of description of the relevant groups in questions (male/female) throughout the manuscript.
---

## [Editor Report · Acceptance letter]

20 Dec 2024

PONE-D-24-38457R2 

PLOS ONE

Dear Dr. Fernandez-Guzman, 

I'm pleased to inform you that your manuscript has been deemed suitable for publication in PLOS ONE. Congratulations! Your manuscript is now being handed over to our production team.

Kind regards, 

on behalf of

Dr. Barry Kweh 

Academic Editor

PLOS ONE